

# 20 Years of ClO Measurements in the Antarctic Lower Stratosphere

Gerald E. Nedoluha[1], Brian J. Connor[2], Thomas Mooney[2], James W. Barrett[3], Alan Parrish[4], R. Michael Gomez[1], Ian Boyd[2], Doug R. Allen[1], Michael Kotkamp[5], Stefanie Kremser[6], Terry Deshler[7], Paul Newman[8], and Michelle L. Santee[9]

[1]Naval Research Laboratory, Washington, D. C., USA

[2]BC Scientific Consulting LLC, Stony Brook, NY, USA

[3]Stony Brook University, Stony Brook, New York, USA

[4]Department of Astronomy, University of Massachusetts, Amherst, MA, USA

[5]National Institute of Water and Atmospheric Research, Lauder, New Zealand

[6]Bodeker Scientific, Alexandra, New Zealand

[7]Department of Atmospheric Science, University of Wyoming, Laramie, WY, USA

[8]NASA Goddard Space Flight Center, Greenbelt, MD, USA

[9]Jet Propulsion Laboratory, California Institute of Technology, Pasadena, California, USA

*Correspondence to:* Gerald E. Nedoluha (nedoluha@nrl.navy.mil)

**Abstract.** We present 20 years of springtime measurements of ClO over Antarctica from the Chlorine monOxide Experiment (ChlOE1) ground-based millimeter wave spectrometer at Scott Base, Antarctica, as well 12 years of ClO measurements from the Aura Microwave Limb Sounder (MLS). From August onwards we observe a strong increase in lower stratospheric ClO, with a peak column amount usually occurring in early September. From mid-September onwards we observe a strong decrease in ClO. In order to study interannual differences we focus on a 3-week period from August 28 to September 17 for each year, and compare the average column ClO anomalies. These column ClO anomalies are shown to be highly correlated with the average ozone mass deficit for September and October of each year. We also show that anomalies in column ClO are anti-correlated with 30 hPa temperature anomalies, both on a daily and an interannual timescale. We calculate the dependence of interannual variations in column ClO on interannual variations in temperature. By making use of this relationship we can better estimate the underlying trend in the Cly which provides the reservoir for the ClO. The resultant trends for zonal MLS, Scott Base MLS (both 2004-2015), and ChlOE (1996-2015) were -0.5±0.2% yr⁻¹, -1.4±0.9% yr⁻¹, and -0.6±0.4% yr⁻¹, respectively. These trends are within 1σ of





trends in stratospheric Cly previously found at other latitudes. This decrease in ClO is the result of changes in anthropogenic CFC emissions due to actions taken under the Montreal Protocol.

## 1. Introduction

Chlorine monoxide (ClO) is central to the formation of the Antarctic ozone hole. It is both the direct product of the reaction between Cl and ozone and the catalytic agent in the most important ozone-depleting chemical cycle (Waters et al., 1993, Salawitch et al., 1993). The Chlorine monOxide Experiment (ChlOE1) ground-based millimeter wave spectrometer was deployed at Scott Base, Antarctica (77.85º S, 166.77º E), by Stony Brook University and the National Institute of Water and Atmospheric Research (NIWA) in February 1996, and is currently jointly operated by the Naval Research Laboratory (NRL) and NIWA. Both this instrument and the ChlOE3 instrument, which operated at Mauna Kea until 2015, are part of the Network for the Detection of Atmospheric Composition Change (NDACC). A new ChlOE4 instrument is being deployed at Mauna Loa in 2016.

In this paper, we describe the measurement technique and present results from the ChlOE1 time series from 1996-2015. Measurements are only shown from mid-August to mid-October, when ClO daytime mixing ratios can reach up to ~2 ppbv. We will also show, from 2004 onwards, ClO measurements from the Aura Microwave Limb Sounder (MLS), both coincident with Scott Base and zonally averaged at the latitude of Scott Base. The ChlOE1 measurements were previously compared with the v1.5 MLS retrievals for the austral spring of 2005 (Connor et al., 2007).

We also show annual anomalies for measurements during the 3 weeks when the stratospheric ClO column densities generally reach their maximum values, and compare these anomalies with interannual anomalies in temperature, as provided by the Modern Era Reanalysis for Research and Applications (MERRA) (Rienecker et al., 2011). We use the 20 years of ChlOE measurements and 12 years of MLS measurements to derive a relationship between the interannual anomalies in ClO column and those in 30 hPa temperature. We then make use of this relationship to derive an estimate of Cly trends in the Antarctic vortex.

## 2. ClO Measurements



The ChlOE ground-based radiometer measures the thermally excited rotational emission lines near 278.63 GHz. The spectrometer bandwidth permits measurement of the pressure-broadened lineshape from which ClO altitude profiles are retrieved. The instrument is a cryogenically cooled (~20 K) heterodyne receiver, tuned to observe the ClO transition by adjustment of a phase-locked local oscillator. It is coupled to a spectrometer with 506 MHz total bandwidth, which is approximately the width of the ClO line at 15 km altitude.

At night, the ClO emission is much weaker and narrower, because nearly all chemically active chlorine ('ClOx') in the lower stratosphere rapidly converts to $Cl_2O_2$ after sunset (Solomon et al., 2002). This allows us, in the ground-based measurements, to remove the instrumental baseline and a small number of interfering atmospheric spectral lines in the instrument bandpass (primarily the ozone line at 278.521 GHz), by subtracting the nighttime spectrum from the daytime one. For these measurements we have defined day as the period from 3 hours after sunrise to 1 hour before sunset, and night as the period from 4 hours after sunset to 1 hour before sunrise. Sunrise and sunset are defined to occur when the solar zenith angle is -94.5°.

A retrieval of the ''day minus night'' spectrum, and thus of the day ClO mixing ratio less the night mixing ratio, is performed by a three-stage process, described in detail by Solomon et al. (2000). The first stage determines the altitude of the peak of the lower stratospheric distribution as a function of date, by performing retrievals on a full season of data, using an a priori profile without a separate lower stratospheric component. In the second stage, the a priori ClO distribution consists of a climatological profile having a peak in the lower stratosphere determined by stage 1 at ~30 hPa (22 km) in mid August to ~48 hPa (19 km) by late September, with a secondary peak in the upper stratosphere at ~5 hPa. The second-stage retrieval is simply a nonlinear least-squares-fit of a single multiplier applied to the lower stratospheric distribution. The climatological distribution, modified by the retrieved multiplier, is used as the a priori distribution for the third stage, which is a maximum a posteriori solution as given by Rodgers (2000, e.g., equation (4.5)). Figure 1 shows a ChlOE1 day minus night retrieval for Sept. 4, 2011. The retrieved and a priori profiles both have two mixing ratio peaks, one in the upper stratosphere and a much larger one in the lower stratosphere. The lower stratospheric peak is only present when inactive chlorine is converted to active chlorine on the surface of polar stratospheric clouds (PSCs), and is therefore only observed under extremely cold Arctic and



Antarctic conditions. The upper stratospheric ClO peak can be observed at any location, but because of the weak signal the best ground-based measurements require extended integrations (~1 week) from a high-altitude site. Measurements of this ClO peak as made from the ChlOE3 instrument at Mauna Kea have been shown in Nedoluha et al. (2011) and Connor et al. (2013).

5   Since all of the measurements shown here will be with the ChlOE1 instrument we will henceforth refer to this instrument simply as ChlOE.

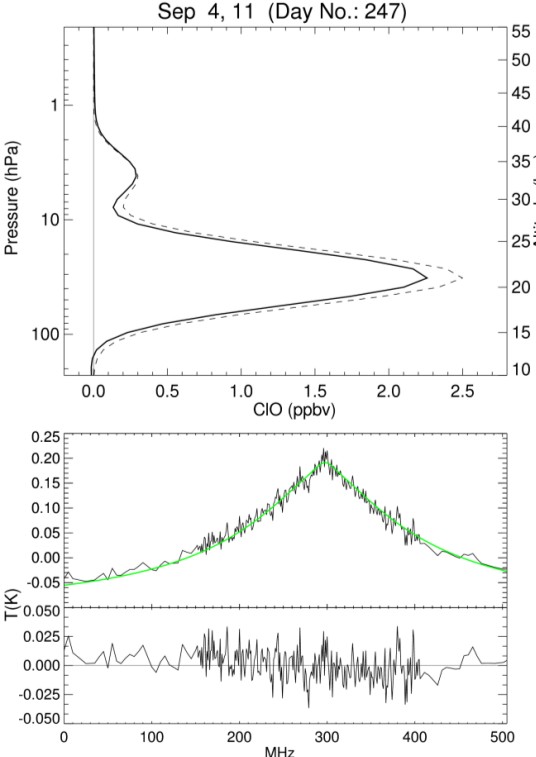

**Figure 1 – Top panel: The retrieved day minus night ClO mixing ratio profile for September 4, 2011 (solid line), and the a priori profile for that day (dashed line). Middle panel: The measured (black line) and modeled (green line) spectra. Bottom panel: The measured-model residual spectrum.**

As is clear in Figure 1, and has previously been shown by ChlOE (Solomon et al, 1987, 2000) and by satellite measurements (e.g., Waters et al, 1993, Santee et al 2005), ClO in the Antarctic spring is overwhelmingly concentrated in the lower stratosphere. We shall, throughout

15   this study, make use of the column ClO at altitudes above 100 hPa. Any variations in this ClO column during the polar PSC season are dominated by changes in the lower stratospheric peak of ClO.





Aura MLS measurements of ClO are available since 2004. Here we use the v4.2 retrievals (Livesey et al., 2015). The Aura measurement overpasses near the latitude of Scott Base occur at ~1630 and ~2300 LST. The times for these measurements remain consistent within ±4 minutes throughout the entire Aura mission. Although it is not possible to replicate

the ChlOE diurnal sampling with the twice daily MLS overpass sampling, we shall nevertheless in this study show exclusively MLS daytime (~1630 LST) minus nighttime (~2300 LST) measurements. We note that at the 78°S latitude of Scott Base the sun actually sets before 1630 at 23 km (i.e., near the ClO peak) until August 24. Figure 2 shows the zonal average day minus night MLS measurements within ±2° latitude of Scott Base for 2006. The MLS measurements

show both the upper and lower stratospheric peaks, with the lower-altitude ClO showing a gradual increase until mid-September, and then a sharp decline.

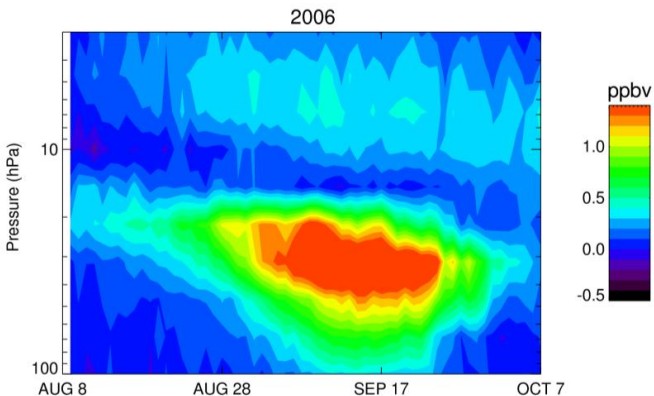

**Figure 2 – The zonally averaged daily ClO mixing ratio (day minus night) as measured by Aura MLS for 2006.**

In Figure 3 we show, for 2006, measurements of day minus night ClO column from ChlOE together with those from the coincident MLS (within ±2° latitude and ±15° longitude of Scott Base) measurements. ChlOE measurements are missing for some days because poor tropospheric weather made it impossible to obtain both the daytime and nighttime spectra required for the retrieval, but the general temporal development is clear in both the MLS and

ChlOE datasets. Essentially there is an increase in August, a maximum in mid-September, and then a rapid decrease at the end of winter.

In addition to the daily measurements for 2006, we also plot the ChlOE and MLS climatologies for these datasets. The ChlOE climatology is derived from measurements taken from 1996-2015, while the MLS climatology is derived from measurements taken from 2004-



2015. Both datasets show values of the 2006 ClO column that, at least until mid-September, are generally higher than their climatologies. Santee et al. (2011) previously noted that this Antarctic winter showed strong and prolonged chlorine activation in the lowermost vortex, and postulated that this was the cause of unusually low column ozone that year.

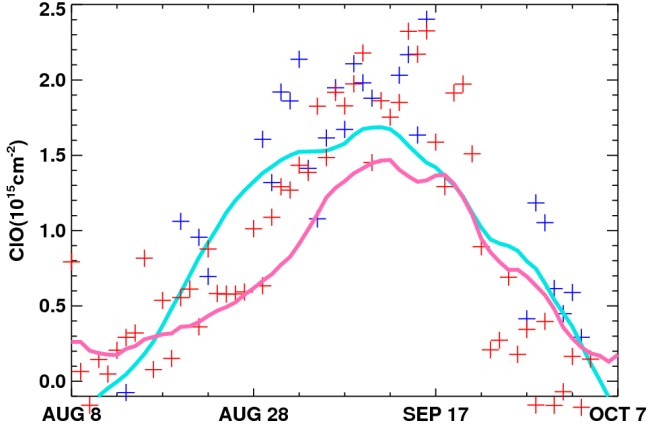

**Figure 3 – Daily (day minus night) column density of ClO measurements at altitudes above 100 hPa for mid-August to mid-October 2006 from ChlOE1 (blue crosses) and from MLS (red crosses). Also shown are climatologies for this period based on the ChlOE measurements from 1996-2015 (light blue line) and MLS measurements from 2004-2015 (pink line).**

Figure 3 shows a clear difference in the seasonal development of the day minus night MLS and ChlOE climatologies. Since the MLS measurements in the vicinity of Scott Base only begin to see sunlit air near the altitude of the ClO peak near August 24, the fast increase in ClO measured by MLS between August 28 and September 7 is to some extent caused by the very

15  large fractional increase in sunlight exposure during this period. Figure 4 shows the diurnal variation of ChlOE ClO column density for measurements at Scott Base on days when hourly measurements were possible. As is seen in the climatologies in Figure 3, the difference between the MLS and ChlOE measurements decreases as the length of daylight increases and the 1630 LST MLS measurement becomes more representative of a mid-day measurement.

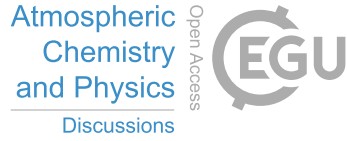

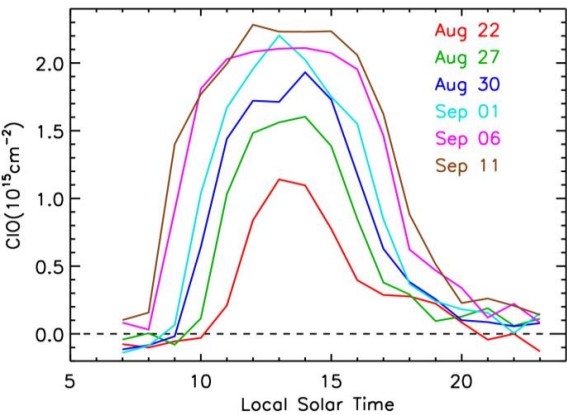

**Figure 4 – The diurnal variation of ClO column density at altitudes above 100 hPa at the Scott Base as measured from a series of measurement days in 2005.  The date given for each curve is the middle date of a 3-day average of hourly measurements.**

We note that Upper Atmosphere Research Satellite (UARS) MLS ClO measurements are available for the years 1991-1993.  Using the ground-based ChlOE measurements from Mauna Kea, we previously showed that the UARS MLS ClO measurements in the upper stratosphere were consistent with the Aura MLS ClO measurements (Nedoluha et al., 2011).  However, unlike Aura MLS, UARS MLS was in a precessing orbit, and therefore the local solar times of it

measurements varied from day to day.  Given the large diurnal variability of lower stratospheric ClO in the vortex we would require an extremely accurate model in order to be able to usefully compare the UARS MLS ClO measurements with other measurements in this study.

**3. Annual ClO Anomalies**

The primary goal of this study is to estimate the trend in Cly near the latitude of Scott Base.  To calculate this trend, we need a period over which we have an adequate number of elevated ClO measurements, and where the year-to-year differences resulting from meteorological variations are minimized.  As was shown in Figures 2 and 3, there is a gradual increase in ClO at the latitude of Scott Base from ~August 28 to September 17.  At the latitude of

Scott Base (77.85°S) we do not expect that during these dates any of the measurements have occurred outside of the vortex (except possibly in 2002, when the Antarctic stratosphere exhibited an unusual major warming).  In the weeks after September 17, there is generally a very sharp drop in ClO, with the exact timing of this drop differing from year to year.  As a measure of this increased variation, we note that the standard deviation of the 12 years of MLS





measurements near Scott Base increases from 29% of the ClO column on September 17 to 58% of the ClO column on September 22. We therefore choose September 17 as the final day of the period for which we will compare interannual variations. On August 28, the climatological ClO from the ChlOE measurements is similar to that on September 17, and the ChlOE measurements

show a steep increase up to this date. The choice of August 28 as the first day for the comparisons gives us a 3-week period that provides an average of 16.4 daily measurements from ChlOE for each year.

We calculate, for the chosen 3-week period, an annual anomaly for each measurement dataset by taking the average difference from the climatology. We then add back on the

climatological average column ClO for the period. So for each year we plot in Figure 5

$$Y(year) = Y_{climo} + [1/n(year)] \Sigma [D(year,day) - D_{climo}(day)] \qquad (1)$$

where $Y_{climo}$ is the dataset-specific climatological average column ClO for the 3-week period,

$n(year)$ is the number of measurements during that period for a specific year, $D(year,day)$ is the measured column ClO for that day, and $D_{climo}(day)$ is the dataset specific climatological average column ClO for that day of the year. MLS anomalies are shown both for the zonal average (within 77.85ºS±2º) and with a further restriction to within ±15º longitude of Scott Base. As expected from the difference in diurnal sampling, the MLS average column day minus night ClO

values are somewhat smaller than those from the ChlOE measurements.

We also show in Figure 5 the average ozone mass deficit (in $10^9$ kg of ozone relative to the 220 DU value) for September and October of each year. This data was obtained from NASA Ozone Watch (ozonewatch.gsfc.nasa.gov) and is based upon data from the Total Ozone Mapping Spectrometer (TOMS) and the Ozone Monitoring Instrument (OMI), with missing data filled in

by the Goddard Earth Observing System Model (GEOS-5).





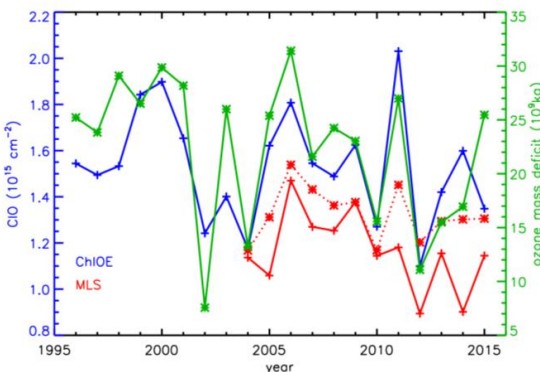

**Figure 5 – The calculated annual average ClO column density for August 28 to September 17 of each year, calculated as described in text. Averages are shown for ChlOE at Scott Base (blue), MLS coincident with Scott Base (solid red) and MLS at the latitude of Scott Base (dashed red line). Also shown is the ozone mass deficit in $10^9$ kg of ozone relative to the 220 DU value (green line, with right-hand axis).**

Figure 5 shows a strong correlation between annual ClO column anomalies and the ozone mass deficit. For the 20 years of ChlOE measurements the correlation coefficient between these is 0.75. The correlation coefficient increases to 0.78 if we do not include 2014, for which there are only 6 ChlOE measurement days out of a possible 21 between August 28 and September 17, as opposed to the annual average of 16.4 measurement days. For the 12 years of Aura MLS ClO column measurements the correlation coefficient is 0.66 for the measurements coincident with Scott Base, and 0.85 for the zonally averaged measurements at the Scott Base latitude.

## 4. Temperature and ClO

The seasonal development of ClO is governed by heterogeneous processes that convert unreactive chlorine species into reactive forms ($ClO_x$). These heterogeneous processes are governed by both temperature and available sunlight. The 3D Single Layer Isentropic Model of Chemistry And Transport (SLIMCAT) model has been used in several studies to compare with ChlOE measurements for specific years (e.g., Solomon et al., 2002).

The fraction of Cly which is in the form of ClO is sensitive to the availability of PSCs, which require low temperatures. As was noted by Santee et al. (2011) for the 2006 winter, the chlorine deactivation and the dissipation of PSCs as observed by CALIPSO (Pitts et al., 2009) both occur in mid-October. While the variation in ClO measured at any one place and time is dependent not just upon the local temperature but upon the temperature history of the measured



parcel, we do find that in many cases sudden changes in local temperature anomalies are quite well correlated with changes in measured ClO. An example of the sensitivity of ClO to changes in temperature is seen very clearly in Figure 1 of Kremser et al. (2011), where a sudden increase in temperature in early September over Scott Base resulted in a sudden decrease in measured

ClO.

Figure 6 shows daily (day minus night) ChlOE column measurements for mid-August to mid-October 2000. This is the same as Figure 3, but for a different (pre-Aura MLS) year. Also shown in Figure 6 are MERRA temperatures at 30 hPa (the pressure level nearest to the ClO mixing ratio peak) within ±2º latitude and ±15º longitude of Scott Base. In addition, we show

the climatological temperature from the same 20-year time period as the ChlOE measurements. While the temperatures in late August are clearly colder than in September, the ClO during this period remains low because of a lack of the sunlight required for the activation of chlorine. Once sunlight becomes available, the ClO column begins to increase, and, in this particular year, increases to levels well above the climatology. As Figure 5 shows, this year is second only to

2011 in the annual average ClO column anomaly for August 28 to September 17. At the same time, Figure 6 shows that the temperatures are colder than the climatology from August 28 to September 15, and are then warmer than the climatology for 6 of the next 7 days. The date when the temperature crosses from below to above the climatological value is the same date on which the ClO column density crosses from above to below the climatology.



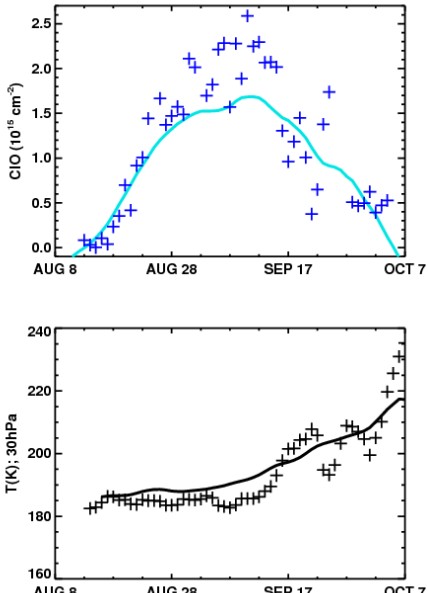

**Figure 6 – Top plot: Daily (day minus night) ChlOE1 column density measurements for mid-August to mid-October 2000 (crosses) and a climatology for that period based on the ChlOE measurements from 1996-2015 (solid line). Bottom plot: Daily 30 hPa temperature measurement from MERRA within ±2º latitude and ±15º longitude of Scott Base (crosses) and a 1996-2015 climatology for this location (solid line).**

The annual average temperature anomalies at three pressure levels (20, 30, and 40 hPa) are plotted in Figure 7. As in Figure 5, these are calculated by taking the average difference between the daily temperature and the temperature climatology for that day over the 3-week period of August 28 to September 17. We find that the relationship between temperatures at

10 these three levels changed between 1998 and 1999. The 20 hPa and 30 hPa temperature anomalies suggested extremely cold years from 1996-1998 (at 20 hPa 1996 and 1997 were the coldest years), while at 40 hPa none of these three years was the coldest in the 20-year record.

In between the 1998 and 1999 periods that we analyzed, data from the Advanced TIROS Operational Vertical Sounder (ATOVS, on NOAA15) began to be introduced into the MERRA,

and it has been shown that this causes some inhomogeneities in the reanalysis (Pawson, 2012). We therefore compared the MERRA temperatures with sondes launched from Scott Base from 1996-1998, and with sondes launched from 1999-2010. We found that the cold bias of MERRA relative to the sondes at 20 hPa was much reduced in the 1999-2010 MERRA temperatures. Based on the biases in MERRA temperatures indicated by the sonde data we added 4.0K to the

20 hPa 1996-1998 MERRA temperatures and 2.1K to the 30 hPa temperatures. We also





subtracted 0.3K from the 40 hPa 1996-1998 MERRA temperatures. When we estimate chlorine

trends in Section 5 it will be particularly important to have temperatures during these first three

years of ChlOE measurements that are consistent with temperatures in later years.

Given the anti-correlation between column ClO and temperature, we would expect an

anti-correlation between temperature and ozone loss. Just as in Figure 5, we therefore also show

in Figure 7 the ozone mass deficit, although in this case with the scale inverted. The correlation,

or more properly, anti-correlation, between the 30 hPa temperature anomalies over Scott Base

and the ozone mass deficit shown in Figure 7 is comparable to that between ClO and ozone mass

deficit shown in Figure 5, with a correlation coefficient of -0.82, while for the zonal average

temperatures the correlation drops to -0.78. The temperature and ozone mass deficit have a

slightly weaker correlation at 40 hPa, while at 20 hPa the correlation is slightly stronger for the

local temperatures and slightly weaker for the zonal average temperatures.

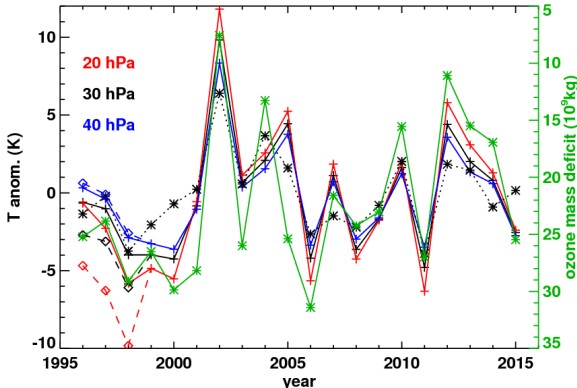

**Figure 7 – Annual average temperature anomalies for August 28 to September 17 within ±2º latitude and ±15º longitude**
**of Scott Base. Results are shown at 20 hPa (red), 30 hPa (black), and 40 hPa (blue). The dashed lines for 1996-1998 show**
**the anomalies before bias correction (see text). Also shown (dotted line black line) is the zonal temperature anomaly for**
**this latitude range. The green line shows the ozone mass deficit, with values shown on the right-hand axis (as in Figure 5,**
**but with the axis reversed).**

Figure 8 presents scatter plots of the annual average column ClO and the 30 hPa

temperature anomalies for August 28 to September 17. Since, after the bias correction, the

temperature anomalies are very similar for all three pressure levels, the anomalies shown in

Figure 8 are nearly independent of the pressure level chosen for the temperatures. We chose 30

hPa since, among the three pressure levels shown in Figure 7, zonally averaged temperatures at

this level showed the highest correlation with MLS zonally averaged column ClO measurements.

For the local MLS measurements and ChlOE, the correlation with local temperatures was



slightly higher at 20 hPa. Results are shown for ChlOE measurements, as well as for MLS measurements both zonally averaged (with corresponding zonally averaged temperatures) and restricted to within ±2º latitude and to within ±15º longitude of Scott Base.

To establish a linear fit for the annual average anomalies shown in Figure 8, we need to estimate uncertainties in the temperature and ClO measurements. We estimate these uncertainties by calculating the standard error of the mean for the daily anomalies for each year. This will tend to weight years that have consistently high (or low) ClO column and temperature anomalies, as well as, for ChlOE, years when there are a large number of measurements (MLS almost always has measurements for every day). The errors for each year are generally similar, but in 2014 there were very few ChlOE measurements and these measurements were particularly variable.

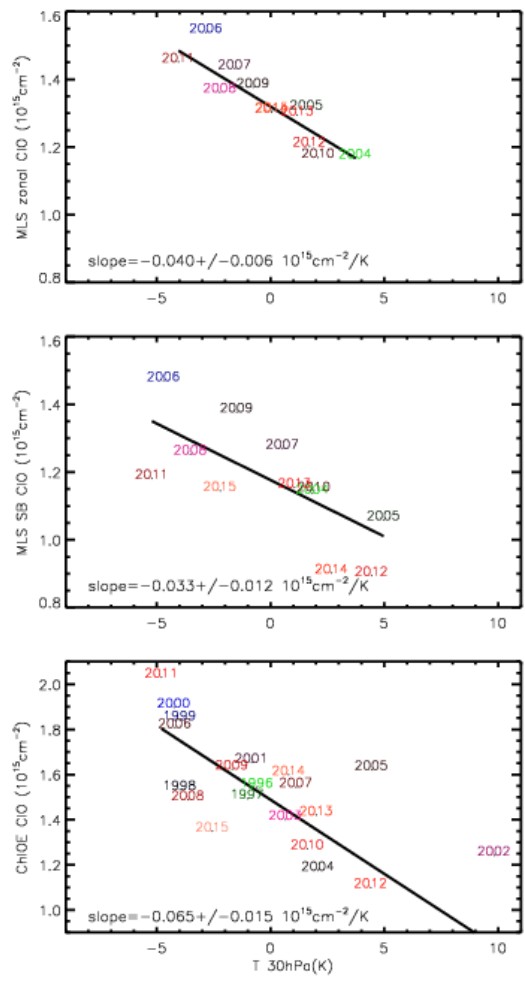

**Figure 8 – The annual average August 28 to September 17 column ClO anomalies (shown in Figure 5) plotted against the temperature anomalies (shown in Figure 7). Also shown are linear fits with a 1σ error estimate. Results are shown for the zonally average MLS ClO column measurements and 30 hPa MERRA temperatures within ±2º latitude of Scott Base (top), for MLS ClO and MERRA temperature with a further restriction to within ±15º longitude (middle) of Scott Base, and for ChlOE ClO measurements and MERRA temperatures with this tighter restriction (bottom).**

We calculated linear fits and found that the slopes were -0.040±0.006 $10^{15}$cm$^{-2}$/K, -0.033±0.012 $10^{15}$cm$^{-2}$/K, and -0.065±0.015 $10^{15}$cm$^{-2}$/K for the zonal MLS, Scott Base MLS, and ChlOE ClO measurements, respectively. We attribute the difference in the linear fits between the ChlOE and MLS ClO measurements to the different diurnal sampling of the ChlOE and MLS



measurements. The slopes for the MLS measurements near Scott Base and for the zonally averaged MLS measurements at this latitude are not statistically different.

As a consistency check, we repeated this study using temperatures from the NCEP Reanalysis (REAN2) (Kistler et al., 2001), and calculated fits that were nearly identical to those shown in Figure 8. The slopes were very close to those calculated with MERRA: $-0.040\pm0.006$ $10^{15}$cm$^{-2}$/K, $-0.032\pm0.012$ $10^{15}$cm$^{-2}$/K, and $-0.068\pm0.015$ $10^{15}$cm$^{-2}$/K for the zonal MLS, Scott Base MLS, and ChlOE ClO measurements, respectively.

## 5. Estimating a Chlorine Trend

There have been a number of studies attempting to quantify the temporal trend in Cly using measurements of either HCl or ClO. HCl is the reservoir species for chlorine, and measurements of HCl in the upper stratosphere show a decline since 1997 (Anderson et al., 2000; Froidevaux et al. (2006); Jones et al., 2011; Nedoluha et al., 2011). Jones et al. (2011) showed, for a range of latitude bands, a decrease in HCl measured by HALOE of between 0.4 and 0.6% yr$^{-1}$ from 1997-2005. Froidevaux et al. (2006) estimated a decrease of $\sim0.8\pm0.1\%$ yr$^{-1}$ from Aura MLS HCl measurements over a very brief August 2004 to January 2006 period, but unfortunately the MLS channel measuring HCl near the stratopause experienced rapid deterioration so no extended HCl trend study from the MLS dataset has been possible. Jones et al. (2011) produced a combined ODIN/SMR and MLS ClO dataset for 2001-2008, and calculated a trend of $-0.7\pm0.8\%$ yr$^{-1}$ ($2\sigma$) in tropical ClO from 35-45km. Nedoluha et al. (2011) used ground-based measurements of ClO from Mauna Kea to show a clear decrease since 1996, and validated the relative consistency of the UARS MLS (1991-1998) and Aura MLS (2004-present) ClO measurements. Finally, Connor et al. (2013) used a reanalyzed version of the ClO measurements from Mauna Kea and calculated a trend of $-0.64\pm0.15\%$ yr$^{-1}$ ($2\sigma$) from 1995-2012.

The linear trend in the annual August 28 to September 7 ChlOE ClO anomalies (those shown in Figure 5) is $-1.1\pm0.4\%$ yr$^{-1}$. However, since the first ChlOE measurement years were colder than average, this trend is almost certainly to some extent the result of increased processing on PSC particles during these years, and is therefore not representative of the trend in Cly. In addition, interannual variations in dynamics will cause interannual variations in Cly which will in turn affect the measured ClO. Strahan et al. (2014) used the compact relationship



between $N_2O$ and Cly, as established by Schauffler et al. (2003), to estimate the variability in Antarctic Cly for the years 2004-2012 based upon MLS measurements of $N_2O$. They found year-to-year variations of Cly in the vortex on the 500K potential temperature surface of as much as ~7%.

Accounting for the dynamical variations over the entire 20-year ClOE measurement dataset is problematic, and we will not attempt to do so here, but it is certainly possible to account for the interannual temperature variations over this period. Making use of the annual temperature anomalies, we calculate an adjusted annual column ClO, which is given for each year by $ClO_{adj}(year) = ClO(year) - \alpha\ \Delta T(year)$, where $\alpha$ is the temperature dependence of ClO

shown in Figure 8 and $\Delta T(year)$ is the temperature anomaly for that year. To the extent that we have successfully removed the effect of temperature variations, these adjusted ClO anomalies should better represent the variation in Cly.

The column ClO anomalies, adjusted for interannual temperature variations, are shown in Figure 9. We then calculate a linear trend using these modified column ClO anomalies and

express the anomaly trends as a function of the average column values. The resultant trends calculated for zonal MLS, Scott Base MLS (2004-2015), and ClOE (1996-2015) are -0.5±0.2% $yr^{-1}$, -1.4±0.9% $yr^{-1}$, and -0.6±0.4% $yr^{-1}$, respectively. Note that the 1σ error bars shown in this plot are the same as the error estimates used in establishing the ClO column vs. temperature relationship in Figure 8. The fraction of points falling within 1σ of the trend line is

approximately what would be expected given a Gaussian distribution, so our uncertainty estimate seems reasonable. If we use the REAN2 temperatures both to establish the relationship between temperature and ClO column and subsequently to calculate trends then we find trends almost identical to those found with the MERRA temperatures. The calculated trends with REAN2 temperatures are -0.6±0.2% $yr^{-1}$, -1.4±0.9% $yr^{-1}$, and -0.5±0.4% $yr^{-1}$ for zonal MLS, Scott Base

MLS, and ClOE.

While the trends are almost insensitive to the choice of temperature dataset, they are somewhat sensitive to the precise choice of dates from which the annual average is determined. Although we believe that we have made an optimal choice for these dates, it is nevertheless instructive to examine this sensitivity. If we add or subtract 5 days from the beginning or end of

the comparison periods and repeat our calculations for these four additional cases we find trends

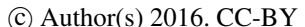



in the range -0.3 to -0.6% yr$^{-1}$ for the zonal MLS measurements, -0.9 to –1.8% yr$^{-1}$ for the local MLS measurements, -0.2 to -0.7% yr$^{-1}$ for the ChlOE measurements.

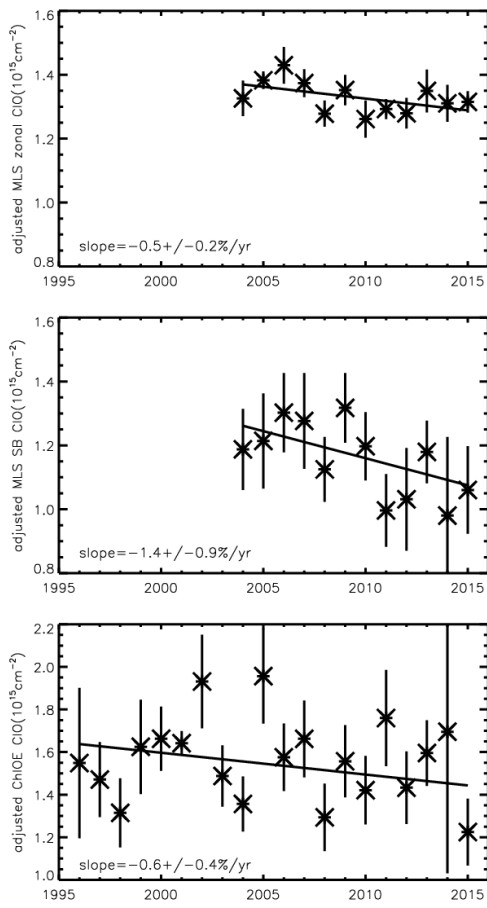

**Figure 9** – **The annual average temperature adjusted ClO anomalies (see text) for August 28 to September 17. The adjustment is based upon the annual average temperature and the relationship shown in Figure 8 (see text). Results are shown for the zonally averaged MLS measurements within ±2º latitude of Scott Base (top), for MLS measurements within ±2º latitude and ±15º longitude of Scott Base, and for ChlOE measurements at Scott Base (bottom). Also shown is a linear fit to the data. Uncertainties are 1σ.**

## 6. Discussion

We have shown column ClO from 20 years of ChlOE measurements over Scott Base, Antarctica, as well as from 12 years of Aura MLS measurements near Scott Base and zonally averaged around 78°S. Interannual variations in column ClO over the 3-week period August 28



to September 17 were correlated with the average ozone mass deficit for September and October ($r$=0.75 for ChlOE). Such a correlation is to be expected, given that ClO is the catalytic agent in the most important ozone-destroying cycle.

We have also shown that the interannual variation in column ClO is anti-correlated with interannual variations in 30 hPa temperature. This is physically reasonable since colder temperatures increase the availability of polar stratospheric clouds, and these will in turn provide the heterogeneous surfaces for the production of ClO (Molina and Molina, 1987; Solomon, 1999).

The multi-year ChlOE and Aura MLS datasets provided the opportunity to study trends. While there have been a number of studies of trends in Cly, this is to our knowledge the first study that addresses the question of stratospheric Cly trends in the Antarctic region. Since the ozone hole represents the most extreme manifestation of ozone depletion, it is of particular interest to determine whether the trends in Antarctic stratospheric Cly, which provides the reservoir for the ClO that causes this destruction, are similar to those measured elsewhere.

Because of the strong dependence of ClO on temperature, any calculated trend in ClO could misrepresent the trend in Cly, particularly if there were unusually warm or cold temperatures near the beginning or end of the timeseries. We therefore used the calculated relationship between interannual variations in column ClO and 30 hPa temperature to account for the effect of variations in column ClO caused by changes in temperature. We then calculated trends in modified ClO. The resultant trends for zonal MLS, Scott Base MLS (2004-2015), and ChlOE (1996-2015) were -0.5±0.2% yr$^{-1}$, -1.4±0.9% yr$^{-1}$, and -0.6±0.4% yr$^{-1}$, respectively.

While our temperature regression does not account for dynamical effects that might influence ClO trends (e.g. changes in the Brewer-Dobson circulation), these trends are within 1σ of trends in Cly previously found at other latitudes (WMO, 2014). This decrease in ClO is the result of changes in anthropogenic CFC emissions due to actions taken under the Montreal Protocol.

## 7. Acknowledgments

This project was funded by NASA under the Upper Atmosphere Research Program, by the Naval Research Laboratory, and by the Office of Naval Research. We would like to acknowledge the many Antarctica New Zealand Technicians who have supported the daily



operation of ChlOE over two of decades measurements. We also acknowledge the logistical support that Antarctica New Zealand has supplied over this period. Work at the Jet Propulsion Laboratory, California Institute of Technology, was carried out under a contract with the National Aeronautics and Space Administration. MLS data are available from the NASA

Goddard Earth Science Data Information and Services Center (acdisc.gsfc.nasa.gov). Sonde temperature data were collected under support from the National Science Foundation.

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
