# Peer review of "Years of ClO Measurements in the Antarctic Lower Stratosphere"

_Atmospheric Chemistry and Physics, 2016_

## Referee Comment (RC1) · Anonymous Referee #1 · 26 Apr 2016

ClO is a tricky molecule to deal with due to its strong diurnal and temperature dependent properties. The authors have done an excellent job in being able to look at Antarctic stratospheric ClO from the ground-based platform (ChlOE) at Scott-Base as well as using satellite data (MLS missions). Analyses using temperature data from MERRA were also performed to account for the relationship between stratospheric ClO and temperature around Scott-Base. As a result of an excellent intercomparison analysis, trends around the Scott-Base region were estimated which were found to be consistent with previous studies as well as providing up to date trends for the Antarctic region.

I recommend this paper for publication after the following revisions are made.

MAJOR P3,L6: There are is one reference to the ClhOE instrument providing some

more detailed information about its specifications and general performance. As it is the one of the main instruments focused upon here in this paper, there should be something written more that helps give the reader an idea of how well it performs. How does it validate to other measurements for example? On P2,L20 it is stated that Connor et al., compared MLS to ClhOE, maybe it would be good to provide a more detailed description in one or two lines of the results here in this paper.

P3, L14. It is stated "For these measurements we have defined day as the period from 3 hours after sunrise to 1 hour before sunset, and night as the period from 4 hours after sunset to 1 hour before sunrise." Is there any particular reason as to why you chose these values? Do the results change significantly if using a larger/smaller range? The text needs some explanation as to why these values were chosen.

Figure 2. I like this plot, but I think there needs to be more explanation for the ClO behavior over time. As I understand it, the small difference between day and night values at the beginning and end of the time period (at both peaks) are because of low ClO activation on a diurnal scale. Is this because there is no sunlight at the beginning of the period, whilst at the end of the period, there is only daylight? I think this needs to be explained a little clearer.

Furthermore, I suggest adding a geometric altitude on the y-axis to stay consistent with Figure 1.

P5, L23: You mention you use a climatology in order to calculate anomaly ClO values. How is this climatology calculated? Do you use a simple mean, median, weighted mean, such as using the uncertainties of the measurements? Did you use all measurements or did you deal with outliers?, if so, how? A little more information is needed here to explain how the climatology was made.

P7, L20: "At the latitude of Scott Base (77.85°S) we do not expect that during these dates any of the measurements have occurred outside of the vortex (except possibly in 2002, when the Antarctic stratosphere exhibited an unusual major warming)".

Is there any way to actually confirm this? There are various tools that can be used to check where the vortex edge is estimated to be situated. One particular tool is Scaled Potential Vorticity (SPV) that has been provided for the MLS data using the GEOS-5 model (Manney et al., 2009).

http://www.atmos-chem-phys.net/9/4775/2009/acp-9-4775-2009.pdf

SPV values are provided for each MLS measurement. As MLS is used here in this paper, you should be able to get a reasonable idea as to what air mass Scott Base was situated in for these measurement periods. If you have possible access to this data, I would suggest trying to confirm this.

Figure 7: There is a good anti-correlation between T and ozone loss, which is seen clearly at most of the maxima and minima here. However, 2004 and 2005 there seems to be a lag between the two, is there any explanation for this?

P15, L12, I think one has to be a little careful concerning the turn-around of 1997. Most previous studies use this year, but it has never really been proven to be the actual turn-around, but best estimate (as far as I'm aware) for the upper stratosphere mid latitudes. I think it would be better to write 'about 1997' or 'around 1997'.

MINOR P1, L17 : I would suggest entering the range of years analyzed here as well stating austral spring. .. "We present 20 years (1996-2015) of austral springtime..". Otherwise, the comments further down about August and September seem somewhat general and could be from any 20 year period.

P1, L22: comma needed. "In order to study inter-annual differences, we focus.."

P1,L22: comma needed. "By making better use of this relationship, we can . . ."

P2, L14: Any reference that could be sited about the future installation at Mauna Loa?

P3, L31: I suggest adding a temperature range/or minimum value for the term 'very cold Antarctic conditions' to which PSCs are formed. According to the WMO, PSCs

form below 195 K.

P4, L2: Comma needed. "because of the weak signal, the best ground-based.."

P4, L3: Commas needed. "Measurements of this ClO peak, as made from the ChlOE3 instrument at Mauna Kea, have been shown in Nedoluha et al. (2011) and Connor et al. (2013)."

P4, L5. Comma needed. "Since all of the measurements shown here 5 will be with the ChlOE1 instrument, we will henceforth refer to this instrument simply as ChlOE".

Figure 1. Caption could read better as; "The retrieved day minus night ClO mixing ratio profile for September 4, 2011 (solid line), and the a priori profile for that day (dashed line), as a function of pressure (left y-axis) and geometric altitude (right y-axis)".

P5, L1: Comma needed. "Here, we use the.."

P5, L8: I would suggest to use pressure as an altitude reference and put the geometric altitude in brackets as a reference. This then stay consistent with the paper up to this point. Hence; whatever pressure is associated with 23 km.

P5, L15: I suggest rewording to; "Figure 3 shows measurements of day minus night ClO column from ChlOE during 2006 together with those from the coincident MLS (within $\pm 2°$ latitude and $\pm 15°$ longitude of Scott Base) measurements."

P7, L9: Spelling; "solar times of its measurements.."

P7, L20: Comma needed; "At the latitude of Scott Base (77.85°S), we do not expect that"

P8, L2: suggest adjusting to; "We therefore choose September 17 as the final day of the period for which we will compare interannual variations as it provides the best representation of the atmospheric state as a result of a lower variance".

P8, L5: Suggest adjusting to; "The choice of 5 August 28 as the first day for the comparisons provides us a 3-week period yielding an average of 16.4 daily measurements from ChIOE for each year."

P9, L3: Are these really anomalies in Fig. 5? Would that not imply fluctuation around zero? I thought you used the anomalies and added them back onto the climatology, hence yielding "the annual average ClO column"?

P9, L13: Comma needed; "For the 12 years of Aura MLS ClO column measurements, the correlation.."

P9, L22: Commas needed; "The fraction of Cly, which is in the form of ClO, is sensitive.."

P9, L23: Also referring to P3, L31, you state "very low temperatures"...A minimum value should be included.

P9, L24: CALIPSO needed to be written in full before abbreviating "(Cloud-Aerosol Lidar and Infrared Pathfinder Satellite Observations"

P10, L2: I would suggest trying to quantify the amount of correlation. How much is "are quite well correlated..". This provides the reader with his/her own interpretation of the relationship between the two variables.

P11, L6: Better suited to read "The annual average MERRA temperature anomalies at three pressure levels..."

P11, L13: What does TIROS stand for? Television InfraRed Operational Sounder? Please write the full name first of the measuring instrument, before abbreviating.

P11, L19: comma needed; "Based on the biases in MERRA temperatures indicated by the sonde data, we added 4.0K to the..."

P11, L19-L20: I would reword this sentence slightly to confirm that the extra K are added to account for biases. "Due to the biases in MERRA temperatures indicated by the sonde data, we added 4.0K to the 20 hPa 1996-1998 MERRA temperatures and

2.1K to the 30 hPa temperatures to account for this offset".

P12, L2: comma needed: "When we estimate chlorine trends in Section 5, it will be particularly important to have..."

P12, L24: Please quantify the correlation value between MLS ClO Z.A. column and zonally average temperature; "showed the highest correlation"

P15, L26: So "The linear trend in the annual August 28 to September 7 ChlOE ClO anomalies (those shown in Figure 5) is -1.1±0.4%/yr" is referring to years 1996 to 2015? Could you please just make that clear. It might be important to the reader as you state trend periods for previous analyses in your literature review.

P16, L25: Please add 'respectively' "The calculated trends with REAN2 temperatures are -0.6±0.2% yr-1, -1.4±0.9% yr-1, and -0.5±0.4% yr-1 for zonal MLS, Scott Base 25 MLS, and ChlOE, respectively".

References: Need some work to make them all consistent with ACP standards, as well as being in chronological order if under the same leading author.

Please also note the supplement to this comment:
http://www.atmos-chem-phys-discuss.net/acp-2016-188/acp-2016-188-RC1-supplement.pdf

---

## Referee Comment (RC2) · Anonymous Referee #2 · 2 Jul 2016

This paper uses long-term observations from a ground-based instrument and satellite to study trends in ClO in the Antarctic lower stratosphere. The ClO observations are adjusted for temperature effects and trends in inorganic chlorine are derived.

Overall, I think that the authors have a powerful dataset but more work is needed before the paper would be acceptable for ACP (i.e. major revisions). I think that they can address these points and a useful paper will result. My comments are summarised below.

Major comments.

1. The aim of the study needs to be clarified. Polar ozone loss is a 'mature' topic and some simple qualitative results are not an advance. It seems that the aim is to use the ClO to derive underlying Cly trends. The motivation for this needs to be made up front.

Why can't we just observe Cly? Why do we need to know what Cly is doing? Is there any implication for detection of polar ozone recovery? Would you expect the same Cly trend in different latitudes (which the abstract implies)?

2. Effect of temperature on ClO. Temperature could affect ClO levels both by changing the extent of chlorine activation (conversion of HCl or ClONO2) or by changing the partitioning of ClOx. There is also the impact of short-term dynamics (vortex movement). The abstract says that ClO is anti-correlated with T both on a daily and interannual timescale. You should explain the mechanisms for this. Specifically, I thought about how T changes might change ClOx partitioning, but T increases would increase ClO and decrease Cl2O2. Daytime ClOx is mostly ClO but there could be a small effect. In any case, it is up the authors to explain (and quantify) how T might affect ClOx so that the temperature correction can be seen to be robust.

3. What about other atmospheric changes contributing to ClO trends? There is a paper by Solomon et al (literally just published) which argues for changes aerosol loading affecting polar ozone loss (and presumably ClO). How big an effect is that?

4. Anti-correlation of ClO and T. Although this is expected qualitatively, I found it interesting how linear this is, especially for the larger scale MLS data. In fact, the scientific interest is not that it occurs but how strong this anti-correlation is. I think that this is what the abstract should emphasise.

Other specific comments

Title: The current title does not give any indication of the scientific message of the paper. This should be modified to indicate what the 20 years of ClO data are used for. . .

Abstract: Lines 15-16. Give the dates covered by the data.

Abstract. Line 29. Define Cly.

Page 2. Line 6. What is the most important cycle? I know it is ClO + ClO in the

Antarctic, but ClO is also involved in ClO + BrO, which is number two. So this is not clear.

Page 2. Section 1. I think this introduction needs a paragraph on the processes involved in polar ozone depletion where you can explain the role of temperature, PSCs, HCl as a reservoir etc. At present little bits of this information is used bit-by-bit in the results and overall it will be confusing to a non-expert.

Page 3. Line 8. Need to define ClOx. The normal definition is Cl + ClO +2Cl2O2, in which case ClO -> Cl2O2 is a repartitioning within ClOx, not a conversion from ClOx.

Page 3. Line 15. Give altitude at which this SZA is sunset.

Page 3. Line 31. Example of information which needs to be in an introductory paragraph on polar processes.

Page 4. Line 4. Where is Mauna Kea? I do know but this is another example of where the authors have not thought about the non-expert readers.

Page 4. Line 5. Change 'will be' to 'are' – that tense fits the paper better.

Page 4. Figure 1. Use (a), (b), (c) for the panels. The way the panels are laid out, it is not clear if there are two or three.

Page 5. Line 24. Does the comparison change if you subsample the ChlOE data to match the MLS time period?

Page 6. Figure 3. There are no error bars or estimate of uncertainty in the figure. What could be added to inform the comparison of the different datasets?

Page 6. Figure 3 caption. You need to say where this plot is for! I.e.the Scott Base station (with latitude details etc). Say that MLS is sampled at station.

Page 7. Line 8. What does 'were consistent' mean quantitatively?

Page 7. Line 15. This is a long way into the paper to state what the primary goal is!

The introduction should state this (and it should be reflected in the title and abstract content).

Page 9. Figure 5 caption. This says both 'annual' and 'each year'. It is not an annual average.

Page 9.Line 10 onwards. It would be helpful to give the correlation coefficients on the plot (with a legend for the lines).

Page 9. Line 17. Another example of background polar chemistry that should be stated earlier. . .

Page 9. Line19. '3D single layer'? I think that SLIMCAT is a 3D model and that SLIMCAT is just a name, not an acronym. In any case, this paragraph does not say anything. What did this studies show which is relevant here?

Page 11. Figure 6 caption. Met. reanalyses should not be classed as 'measurements'.

Page 12. Line 7. 'or more properly' – just say what it really is. Choose one way of saying it.

Page 14. Line 8. Why not reduce power of 10 and remove some decimal places?

Page 17. Line 10. This is a Summary or Conclusions. It is the final section and it does not add any more discussion. The lack of a conclusions section gives the impression that in this draft the authors were clear about their main scientific message.

---

## Author Comment (AC1) · 18 Jul 2016

We have responded to all of the comments provided by both reviewers, which were very helpful in clarifying the manuscript. Our direct responses are indicated in red below. A manuscript with indicated changes has also been submitted.

**Reviewer 1**

ClO is a tricky molecule to deal with due to its strong diurnal and temperature dependent properties. The authors have done an excellent job in being able to look at Antarctic stratospheric ClO from the ground-based platform (ChlOE) at Scott-Base as well as using satellite data (MLS missions). Analyses using temperature data from MERRA were also performed to account for the relationship between stratospheric ClO and temperature around Scott-Base. As a result of an excellent intercomparison analysis, trends around the Scott-Base region were estimated which were found to be consistent with previous studies as well as providing up to date trends for the Antarctic region. I recommend this paper for publication after the following revisions are made.

Thank you for the kind words and the recommendation. With very few exceptions we agree that all of the suggested changes improve the manuscript.

**MAJOR**

**P3,L6:** There are is one reference to the ClhOE instrument providing some more detailed information about its specifications and general performance. As it is the one of the main instruments focused upon here in this paper, there should be something written more that helps give the reader an idea of how well it performs. How does it validate to other measurements for example? On P2,L20 it is stated that Connor et al., compared MLS to ClhOE, maybe it would be good to provide a more detailed description in one or two lines of the results here in this paper.

We have added some more details about the history of ChlOE measurements in the previous paragraph, and have added from the Connor et al. (2007) the result that "comparisons of measurements taken within ±30 minutes of the MLS ascending orbit overpass showed agreement of 11±8% in the peak mixing ratios".

**P3, L14**. It is stated *"For these measurements we have defined day as the period from 3 hours after sunrise to 1 hour before sunset, and night as the period from 4 hours after sunset to 1 hour before sunrise."*

Is there any particular reason as to why you chose these values? Do the results change significantly if using a larger/smaller range? The text needs some explanation as to why these values were chosen.

The values were empirically determined based on avoiding periods of rapid change in ClO. Explanatory text has been added.

**Figure 2**. I like this plot, but I think there needs to be more explanation for the ClO behavior over time. As I understand it, the small difference between day and night values at the beginning and end of the time period (at both peaks) are because of low ClO activation on a diurnal scale. Is this because there is no sunlight at the beginning of the period, whilst at the end of the period, there is only daylight? I think this needs to be explained a little clearer.

We have added some text here regarding the seasonal evolution of the partitioning between ClO, HCl, and ClONO$_2$. We have also clarified that this Figure would look similar if we showed MLS daytime measurements only.

Furthermore, I suggest adding a geometric altitude on the y-axis to stay consistent with Figure 1.

done

**P5, L23**: You mention you use a climatology in order to calculate anomaly ClO values. How is this climatology calculated? Do you use a simple mean, median, weighted mean, such as using the uncertainties of the measurements? Did you use all measurements or did you deal with outliers?, if so, how? A little more information is needed here to explain how the climatology was made.

We have clarified that the climatology is based on simple means with a 5-day smoothing. We do point out that ChlOE measurements are missing on some days due to poor tropospheric weather.

**P7, L20**: "**At the latitude of Scott Base (77.85ºS) we do not expect that during these dates any of the measurements have occurred outside of the vortex (except possibly in 2002, when the Antarctic stratosphere exhibited an unusual major warming)**".

Is there any way to actually confirm this? There are various tools that can be used to check where the vortex edge is estimated to be situated. One particular tool is Scaled Potential Vorticity (SPV) that has been provided for the MLS data using the GEOS-5 model (Manney et al., 2009). http://www.atmos-chem-phys.net/9/4775/2009/acp-9-4775-2009.pdf SPV values are provided for each MLS measurement. As MLS is used here in this paper, you should be able to get a reasonable idea as to what air mass Scott Base was situated in for these measurement periods. If you have possible access to this data, I would suggest trying to confirm this.

Unfortunately there were no MLS measurements in 2002, so the SPV provided for the MLS data does not help us. In other years Scott Base seems to be well within the vortex, but given that there are different vortex-edge definitions we are hesitant to make a definitive statement.

**Figure 7**: There is a good anti-correlation between T and ozone loss, which is seen clearly at most of the maxima and minima here. However, 2004 and 2005 there seems to be a lag between the two, is there any explanation for this?

We are certainly not aware of any physical mechanism which would provide for a lag between the two years.

**P15, L12**, I think one has to be a little careful concerning the turn-around of 1997. Most previous studies use this year, but it has never really been proven to be the actual turn-around, but best estimate (as far as I'm aware) for the upper stratosphere mid latitudes. I think it would be better to write 'about 1997' or 'around 1997'.

We certainly do not wish to indicate anything definitive about 1997. 'Around 1997' is certainly a better way to phrase this.

**MINOR**

**P1, L17** : I would suggest entering the range of years analyzed here as well stating austral spring. .. "We present 20 years (1996-2015) of austral springtime..". Otherwise, the comments further down about August and September seem somewhat general and could be from any 20 year period. done

**P1, L22**: comma needed. "In order to study inter-annual differences, we focus.." done

**P1,L22**: comma needed. "By making better use of this relationship, we can …" done

**P2, L14**: Any reference that could be sited about the future installation at Mauna Loa?

This has been changed to indicate that we now have an instrument operating from Mauna Loa. Since we have just started taking measurement we have nothing to reference.

**P3, L31**: I suggest adding a temperature range/or minimum value for the term 'very cold Antarctic conditions' to which PSCs are formed. According to the WMO, PSCs form below 195 K. done

**P4, L2**: Comma needed. "because of the weak signal, the best ground-based.." done

**P4, L3**: Commas needed. "Measurements of this ClO peak, as made from the ChlOE3 done instrument at Mauna Kea, have been shown in Nedoluha et al. (2011) and Connor et al. (2013)."

**P4, L5**. Comma needed. "Since all of the measurements shown here 5 will be with the ChlOE1 instrument, we will henceforth refer to this instrument simply as ChlOE". done

**Figure 1**. Caption could read better as; "The retrieved day minus night ClO mixing ratio profile for September 4, 2011 (solid line), and the a priori profile for that day (dashed line), as a function of pressure (left y-axis) and geometric altitude (right y-axis)". done

**P5, L1**: Comma needed. "Here, we use the.." done

**P5, L8**: I would suggest to use pressure as an altitude reference and put the geometric altitude in brackets as a reference. This then stay consistent with the paper up to this point. Hence; whatever pressure is associated with 23 km. done

**P5, L15**: I suggest rewording to; "Figure 3 shows measurements of day minus night ClO column from ChlOE during 2006 together with those from the coincident MLS (within ±2º latitude and ±15º longitude of Scott Base) measurements." done

**P7, L9**: Spelling; "solar times of its measurements.." done

**P7, L20**: Comma needed; "At the latitude of Scott Base (77.85ºS), we do not expect that" done

**P8, L2**: suggest adjusting to; "We therefore choose September 17 as the final day of the period for which we will compare interannual variations as it provides the best representation of the atmospheric state as a result of a lower variance".

The previous sentence already notes the increase in variance that occurs if we extend to Sept. 22, so it seems repetitive to say this again.

**P8, L5**: Suggest adjusting to; "The choice of 5 August 28 as the first day for the comparisons provides us a 3-week period yielding an average of 16.4 daily measurements from ChlOE for each year." done

**P9, L3**: Are these really anomalies in Fig. 5? Would that not imply fluctuation around zero? I thought you used the anomalies and added them back onto the climatology, hence yielding "the annual average ClO column"?

The word "anomaly" has been using incorrectly in several places in the manuscript. As the reviewer suggests, we have replaced this with "annual average ClO column", or some similarly appropriate phrase.

**P9, L13**: Comma needed; "For the 12 years of Aura MLS ClO column measurements, the correlation.." done

**P9, L22**: Commas needed; "The fraction of Cly, which is in the form of ClO, is sensitive.."

We changed "which" to "that" without adding commas.

**P9, L23**: Also referring to P3, L31, you state "very low temperatures"…A minimum value should be included. done

**P9, L24**: CALIPSO needed to be written in full before abbreviating "(Cloud-Aerosol Lidar and Infrared Pathfinder Satellite Observations" done

**P10, L2**: I would suggest trying to quantify the amount of correlation. How much is "are quite well correlated..". This provides the reader with his/her own interpretation of the relationship between the two variables.

Since this refers to a number of individual cases, it is difficult to quantify. We changed the wording to read "coincide with".

**P11, L6**: Better suited to read "The annual average MERRA temperature anomalies at three pressure levels…" done

**P11, L13**: What does TIROS stand for? Television InfraRed Operational Sounder? Please write the full name first of the measuring instrument, before abbreviating.
We now provide the full name, Television Infrared Observation Satellite, in the text.

**P11, L19**: comma needed; "Based on the biases in MERRA temperatures indicated by the sonde data, we added 4.0K to the…" done

**P11, L19-L20**: I would reword this sentence slightly to confirm that the extra K are added to account for biases. "Due to the biases in MERRA temperatures indicated by the sonde data, we added 4.0K to the 20 hPa 1996-1998 MERRA temperatures and 2.1K to the 30 hPa temperatures to account for this offset". done

**P12, L2**: comma needed: "When we estimate chlorine trends in Section 5, it will be particularly important to have…" done

**P12, L24**: Please quantify the correlation value between MLS ClO Z.A. column and zonally average temperature; "showed the highest correlation" done

**P15, L26**: So "The linear trend in the annual August 28 to September 7 ChlOE ClO anomalies (those shown in Figure 5) is -1.1±0.4%/yr" is referring to years 1996 to 2015? Could you please just make that clear. It might be important to the reader as you state trend periods for previous analyses in your literature review.
Yes, this is a good point. We have added the years.

**P16, L25**: Please add 'respectively' "The calculated trends with REAN2 temperatures are -0.6±0.2% $yr_{-1}$, -1.4±0.9% $yr_{-1}$, and -0.5±0.4% $yr_{-1}$ for zonal MLS, Scott Base 25 MLS, and ChlOE, respectively". done

**References**: Need some work to make them all consistent with ACP standards, as well as being in chronological order if under the same leading author.

**Reviewer 2**

This paper uses long-term observations from a ground-based instrument and satellite to study trends in ClO in the Antarctic lower stratosphere. The ClO observations are adjusted for temperature effects and trends in inorganic chlorine are derived. Overall, I think that the authors have a powerful dataset but more work is needed before the paper would be acceptable for ACP (i.e. major revisions). I think that they can address these points and a useful paper will result. My comments are summarized below.
The reviewer comments are primarily aimed at improving the presentation to clarify the goals and to provide some relevant background. We hope that our response to the comments adequately addresses these weaknesses in the original manuscript.
Major comments.
1. The aim of the study needs to be clarified. Polar ozone loss is a 'mature' topic and some simple qualitative results are not an advance. It seems that the aim is to use the ClO to derive underlying Cly trends. The motivation for this needs to be made up front.
The introduction has been extended to emphasize that we are studying trends. We would also like to point out that no results from this ground-based dataset have been published for many years, so in part the motivation really is simply to make the community aware of these unique long-term measurements.
Why can't we just observe Cly?
We now state explicitly that ClO has emission lines at microwave frequencies.

Why do we need to know what Cly is doing? Is there any implication for detection of polar ozone recovery?

Text has been added to the introduction to address these questions.

Would you expect the same Cly trend in different latitudes (which the abstract implies)?

Yes, we would expect similar trends in Cly at different latitudes, however it was not obvious how closely the magnitude would agree.

2. Effect of temperature on ClO. Temperature could affect ClO levels both by changing the extent of chlorine activation (conversion of HCl or ClONO2) or by changing the partitioning of ClOx. There is also the impact of short-term dynamics (vortex movement). The abstract says that ClO is anti-correlated with T both on a daily and interannual timescale. You should explain the mechanisms for this. Specifically, I thought about how T changes might change ClOx partitioning, but T increases would increase ClO and decrease Cl2O2. Daytime ClOx is mostly ClO but there could be a small effect. In any case, it is up the authors to explain (and quantify) how T might affect ClOx so that the temperature correction can be seen to be robust.

We have added text, primarily in the introduction, discussing the importance of PSCs and in turn their dependence on temperature. While there are certainly other chemical mechanisms involved, we think that it is the link between PSCs and temperature which is the dominant driver of the anti-correlation between ClO and temperature. However, we have not run any model calculations and are therefore not in a position to present any detailed mechanism. The results shown in the manuscript rely completely on the phenomenological relationship which we calculate from the measurements.

We also reran the entire analysis chain using daytime MLS measurements only, and found that the results were very similar (agreement to within $1\sigma$). We added this result to the text near Figure 2. We also note that if we use nighttime measurements only a very week correlation with temperature (-0.22), and a statistically insignificant slope when comparing to ClO column and temperature as in Figure 8.

3. What about other atmospheric changes contributing to ClO trends? There is a paper by Solomon et al (literally just published) which argues for changes aerosol loading affecting polar ozone loss (and presumably ClO). How big an effect is that?

As Solomon et al. (2016) show, chemistry and dynamic dominates the variations, and it is the dynamical component which we have tried to remove with our temperature correction. They do show that aerosols from volcanoes have some effect on ozone loss, but this is well below our level of sensitivity. Solomon et al. suggest that volcanic aerosols caused a reduction in the "healing trend" of ~10% in ozone, but the error bars associated with our calculated Cly trends are well above 10% of the trend values.

4. Anti-correlation of ClO and T. Although this is expected qualitatively, I found it interesting how linear this is, especially for the larger scale MLS data. In fact, the scientific interest is not that it occurs but how strong this anti-correlation is. I think that this is what the abstract should emphasise.

We have added text to the abstract to emphasize the strong correlation and the calculation of a linear fit.

Other specific comments

Title: The current title does not give any indication of the scientific message of the paper. This should be modified to indicate what the 20 years of ClO data are used for. . .

We have added an emphasis on the implications of these measurements in the paper, but we have not changed the title.

Abstract: Lines 15-16. Give the dates covered by the data. done

Abstract. Line 29. Define Cly. done

Page 2. Line 6. What is the most important cycle? I know it is ClO + ClO in the Antarctic, but ClO is also involved in ClO + BrO, which is number two. So this is not clear.

We now specify the chemical cycle

Page 2. Section 1. I think this introduction needs a paragraph on the processes involved in polar ozone depletion where you can explain the role of temperature, PSCs, HCl as a reservoir etc. At present little bits of this information is used bit-by-bit in the results and overall it will be confusing to a non-expert.

We have added several sentences to the introduction, and have split the first paragraph into two, so that the first paragraph discusses the processes involved in polar ozone depletion, and the second provides information on ClO measurements.

Page 3. Line 8. Need to define ClOx. The normal definition is Cl + ClO +2Cl2O2, in which case ClO -> Cl2O2 is a repartitioning within ClOx, not a conversion from ClOx.

Yes, this was incorrectly expressed. It now reads: "all ClO … converts to Cl2O2".

Page 3. Line 15. Give altitude at which this SZA is sunset. done

Page 3. Line 31. Example of information which needs to be in an introductory paragraph on polar processes. Now in both places.

Page 4. Line 4. Where is Mauna Kea? I do know but this is another example of where the authors have not thought about the non-expert readers. done

Page 4. Line 5. Change 'will be' to 'are' – that tense fits the paper better. done

Page 4. Figure 1. Use (a), (b), (c) for the panels. The way the panels are laid out, it is not clear if there are two or three. done

Page 5. Line 24. Does the comparison change if you subsample the ChlOE data to match the MLS time period?

No, it makes almost no difference. We have now indicated this in the text.

Page 6. Figure 3. There are no error bars or estimate of uncertainty in the figure. What could be added to inform the comparison of the different datasets?

We provide here no error analysis of individual measurements. However, we do make use of the day-to-day scatter for each year (shown in this figure for 2006) to estimate (later in the paper) the uncertainties for the annual averages which are the main focus of this paper.

Page 6. Figure 3 caption. You need to say where this plot is for! I.e.the Scott Base station (with latitude details etc). Say that MLS is sampled at station. done

Page 7. Line 8. What does 'were consistent' mean quantitatively? done

Page 7. Line 15. This is a long way into the paper to state what the primary goal is! The introduction should state this (and it should be reflected in the title and abstract content).

This sentence has been moved to the introduction and a different sentence now introduces this Section.

Page 9. Figure 5 caption. This says both 'annual' and 'each year'. It is not an annual average.

The caption here, and in other places, has been rewritten to say "climatology plus annual average anomally".

Page 9.Line 10 onwards. It would be helpful to give the correlation coefficients on the plot (with a legend for the lines).

The correlation coefficients are now indicated on the plot.

Page 9. Line 17. Another example of background polar chemistry that should be stated earlier. . .

We have added some of this to the now much extended introduction.

Page 9. Line19. '3D single layer'? I think that SLIMCAT is a 3D model and that SLIMCAT is just a name, not an acronym. In any case, this paragraph does not say anything. What did this studies show which is relevant here?

This truly unnecessary paragraph has been eliminated.

Page 11. Figure 6 caption. Met. reanalyses should not be classed as 'measurements'.

done

Page 12. Line 7. 'or more properly' – just say what it really is. Choose one way of saying it.

This awkward wording was introduced because correlations and anti-correlations are being compared. We now write "magnitude of the anti-correlation".

Page 14. Line 8. Why not reduce power of 10 and remove some decimal places?

This notation is used because the measured columns themselves are given in units of $10^{15}\,cm^{-2}$/K.

Page 17. Line 10. This is a Summary or Conclusions. It is the final section and it does not add any more discussion. The lack of a conclusions section gives the impression that in this draft the authors were clear about their main scientific message.

We agree with the reviewer, and the title of this Section has been changed to Summary.

[revised manuscript text omitted]